

# Influence of long-term participation in amateur sports on physical posture of teenagers

Yongchao Huang[1], Meiling Zhai[2], Shi Zhou[3], Yahong Jin[4], Li Wen[5], Yuqi Zhao[2] and Xu Han[2]

[1] Medical College, Zhengzhou University of Industrial Technology, Zhengzhou, Henan, China
[2] Institute of Exercise and Health, Tianjin University of Sport, Tianjin, China
[3] Discipline of Sport and Exercise Science, Southern Cross University, Lismore, Australia
[4] Nanjing Sport Institute, School of Physical Education and Humanities, Nanjing, Jiangsu, China
[5] School of Sports and Health, Nanjing Sport Institute, Nanjing, Jiangshu, China

Corresponding author
Li Wen, wenli34@hotmail.com

## ABSTRACT

**Aim.** The aim of this cross-sectional study was to explore the influence of long-term participating in amateur sports on body posture of school students.

**Methods.** A survey on sport participation was conducted on 1,658 volunteer students aged from 6 to 17 years in two primary schools and one meddle school in Tianjin city. The PA200LE body posture evaluation system and a SpineScanSH-115 electronic spine measuring instrument were used to assess the participants' body posture. According to the survey results and against the inclusion and exclusion criteria, 1,124 eligible participants were divided into seven sport participation groups and seven age-matched control groups for statistical comparisons.

**Results.** Compared with the age-matched controls, the natural standing thoracic kyphosis angle of the swimming group ($35.0 \pm 9.5$ VS $31.2 \pm 8.5$; $t = -2.560$; $p = 0.011$) and the football group ($34.6 \pm 7.2$ VS $31.9 \pm 7.5$; $t = -2.754$; $p = 0.006$) were found to be significantly larger; the natural standing lumbar lordosis angle ($-23.0 \pm 11.0$ VS $-27.0 \pm 11.1$; $t = 0.344$; $p = 0.024$) and the upright sitting lumbar lordosis angle ($-11.7 \pm 8.4$ VS $-15.2 \pm 12.3$; $t = 5.738$; $p = 0.030$) of the swimming group was significantly smaller; the upright sitting thoracic kyphosis angle of the running group ($25.1 \pm 9.3$ VS $22.6 \pm 9.9$; $t = -1.970$; $p = 0.050$) was significantly larger; the upright sitting thoracic kyphosis angle ($23.9 \pm 8.9$ VS $27.0 \pm 7.6$; $t = 2.096$; $p = 0.038$), the learning position thoracic kyphosis angle ($31.0 \pm 8.6$ VS $37.1 \pm 8.9$; $t = 3.076$; $p = 0.003$), the shoulder level ($-1.3 \pm 2.1$ VS $0.0 \pm 2.5$; $t = 2.389$; $p = 0.019$) and waist level ($-1.2 \pm 1.7$ VS $-0.3 \pm 1.7$; $t = 2.511$; $p = 0.013$) of the table tennis group were significantly smaller.

**Conclusions.** The results showed that long-term participation in recreational sports training had an impact on the physical posture of adolescents.

## INTRODUCTION

It is known that long-term participation in certain specific sports training will lead to adaptive changes in body posture due to the physiological load on the musculoskeletal

system and the biomechanical characteristics of the exercise or instrument (*Grabara, 2012*; *Muyor, López-Miñarro & Alacid, 2011*; *Rajabi et al., 2008*; *Alricsson & Werner, 2006*). For example, *Grabara (2012)* analyzed the body posture of 73 Polish footballers and 78 untrained peers, and reported that the young male footballers had smaller lumbar lordosis angles (LLA) than their untrained peers. *Muyor, López-Miñarro & Alacid (2011)* compared the thoracic kyphosis angle (TKA) of 60 professional cyclists and 68 non-cyclists of the same age, and found that the TKA in the professional cyclists was higher than that in the non-cyclists. *Rajabi et al. (2008)* reported that the TKA was the highest in the freestyle wrestlers, followed by non-athletes, and the lowest in the Greco-Roman wrestlers. *Alricsson & Werner (2006)* evaluated the changes of spinal curvature in teenager skiers and reported that the TKA of elite skiers increased significantly after 5 years of training.

The adaptations in the musculoskeletal system to professional sport training is essential to achieve better performance (*Modi et al., 2008*; *Fortin et al., 2012*; *Jones et al., 2005*). However, the repeated exercise in an unnatural posture under the action of static and dynamic loads may attenuate the adaptability of the passive components of the spine, as well as the adaptive ability of the active components, including the muscles that maintain the normal shape of the spine (*Schiller & Eberson, 2008*). These changes could be the reason for the specific morphological characteristics found the well-trained athletes.

It is also known that regular participation in adequate levels of physical exercise and sport activities is beneficial to the health and physical fitness of children and adolescents (*Patel et al., 2021*; *Heydenreich, Schweter & Lührmann, 2020*). The children and adolescents are encouraged to participate in exercise and sports during and out of their school time (*Batez et al., 2021*; *Schmidt et al., 2017*). School-age children and adolescents are in the stage of growth and development, and the adaptions to the exercise and sport training may have life-long impacts on their health and cognitive ability (*Takehara et al., 2019*; *Gomez-Pinilla & Hillman, 2013*). However, there have been few reports on whether participation in regular training of certain sports at amateur levels would have specific impacts on teenagers' body posture.

The exercise frequency, intensity and time in amateur sport training may not match the level of professional athletes. However, the impacts of sport training at this level on the physical posture of teenagers have not been extensively examined. It was hypothesized that long-term participation in amateur sports will affect physical posture in teenagers. Therefore, the purpose of this study was to explore the impacts of long-term participation in a number of popular sports at amateur levels on physical posture, in a convenient sample of school students from Tianjin city, China.

## MATERIALS & METHODS

### Study design
The participants of this study were recruited from two primary schools and one middle school in Tianjin city, China, with a total number of 1,658. This cross-sectional investigation included a survey questionnaire on sports participation, and a number of body posture tests. According to the results of the survey, the participants who meet the inclusion criteria

were divided into an exercise group and a control group. Then the body posture tests were performed for a comparison between the two groups in each type of sport under investigation.

The study was approved by the Ethics Committee of Tianjin University of Sport (approval number TJUS2019032).

## Participants

All the participants and their parents/guardians were informed of the procedure and voluntarily signed the informed consent form before the commencement of data collection. According to a survey on their sports participation, the volunteers were divided into the following seven sport groups (SGs): swimming, table tennis, running, basketball, badminton, football and rope skipping, and their age-matched control groups (CGs). The inclusion criteria for the SGs were: the time of self-practice or training under the guidance of club coach was no less than 4 h per week and the training history was no less than two years. If a participant was only engaged in one sport for two years or longer, the participant was assigned to that specific SG. If the participants took part in a certain sport for less than one hour every week and the training length was less than one year, they were assigned to the CGs.

Initially, 1,658 students volunteered to participate in the study. To be able to clearly identify the effects of long-term sports training in this cross-sectional study, students were excluded from the study if their sport training time was between the above-defined two groups (*i.e.,* between one and two years); they were engaged in multiple sports simultaneously; they participated in regular physical activities other than sports; or with traumatic back injury, scoliosis more than 20° (Cobb angle) or documented congenital spinal malformation. Against the exclusion criteria, 534 volunteers were excluded because their sport training time was between one and two years ($n = 325$), engaged in multiple sports simultaneously ($n = 176$), or suffered from traumatic back injury ($n = 33$). These exclusion criteria were applied aiming to make the characteristics more distinguishable between the SGs and the CGs. Therefore, a total of 1124 eligible participants were included in further date collection and analysis in this study.

## Procedure

A purpose-made survey questionnaire was used to collect information about personal sport participation history, musculoskeletal injury and self-reported training plan among the volunteer students. The questionnaire was completed by students themselves.

In order to verify the reliability of the data collection in the survey, 50 participants were randomly selected from the sample pool, and the questionnaire was completed three times by each participant. The second completion was conducted 24 h after the first one, and the third completion took place two weeks after the first one. Then, Kappa statistical analysis was performed to determine the level of reliability. To evaluate the reliability of the body posture measurements, the same 50 participants as used in the above-mentioned questionnaire reliability evaluation were measured twice by the same researcher, and the second measurement took place 24 h after the first one. Then, the intra-group correlation coefficient (ICC) was used to test the consistency of the repeated measurements.

Height and weight. Standard stadiometers and scales were used to measure the body height and weight. During the measurement, the participant stood barefoot, wearing only close-fitting attire, standing in an upright position on the platform of the stadiometer that was also a weighing scale, torso straight, the arms dropping naturally, heels together, toes apart about 60 degrees, with the heel, sacrum and shoulder blades in contact with the height column, head upright, and eyes looking forward. The head position was adjusted so that an imaginary line between the lowest point the superior edge of the tragus and the inferior margin of the orbit was horizontal. The accuracy of the height measurement was to the nearest one mm, and the accuracy of the weight measurement was to 10 g. A total of three measurements were made, and the average values of three measurements were used in further analysis.

Posture assessments. A PA200LE Body Monitoring and Analysis System (The Big Sports Co. Ltd., Japan) was utilized to assess the neck inclination (NI), pelvic inclination (PI), forward neck (FN), forward upper part of body (FUPB), forward lower part of the body (FLPB), neck torsion (NT), shoulder torsion (ST), waist torsion (WT), shoulder level (SL), waist level (WL), lateral neck inclination (LNI), lateral upper part of body inclination (LUPBI) and lateral lower part of body inclination (LLPBI). In a private room, the participant stood on the plantar pressure plate, with the camera located 2.4 m in front of the plate, and the red cross bean of the camera was calibrated to be coincided with the white cross on the plantar pressure plate before the commencement of assessment. During the postural assessments, the participants wore close-fitting attire, the markers for photo-image analysis were affixed to the anatomic landmarks on the body. The landmarks included the sternal notch, the spinous process of the seventh cervical vertebra, the greater tubercle of the left and right humerus, the navel, the left and right posterior superior iliac spine, the left and right anterior superior iliac spine, the left and right greater trochanter of femur, the lateral condyle of tibia of the left and right legs, the left and right patella, and the fifth trochanter of the left and right foot. The participants stood on the plantar pressure plate with their feet 10 cm distance apart and the fifth metatarsal tuberosity of their feet aligned with the white line in the sagittal direction of the plantar pressure plate, looking forward, and their hands dropping naturally on both sides of the body. While keeping the left and right markers of the fifth metatarsal on the platform line, the participant was instructed to make four 90-degree rotations and obtained four photos of the front, sides and back views. A total of three measurements were made for each participant, and the average values of three measurements was used in further analysis.

The TKA and LLA were measured in three postures, including natural standing, upright sitting and desk learning postures, using the electronic spine measuring instrument SpineScanSH-115(The Sunlight Co. Ltd., Israel). During the measurement, the participant first stood in a natural posture, the electronic spine measuring instrument was applied along the spine from C7 to S1, to measure the NSTKA and the NSLLA. Then the participant was asked to sit upright in the chair similar to what they used in the school, to measure the USTKA and the upright sitting lumbar lordosis angle (USLLA). The participant was then asked to copy a passage in their normal desk learning position for 5 min, to measure the

LTKA and the learning lumbar lordosis angle (LLLA). A total of three measurements were made, and the average value was used in further analysis.

Definitions of the measured variables:

TKA: the angle between T2 upper endplate and T12 lower endplate.

LLA: the angle between the L1 superior endplate and the L5 inferior endplate.

NI: The angle between the ear foramen, the spinous process of the seventh cervical vertebra and the greater tubercle of the humerus.

PI: The angle between the line between the anterior superior iliac spine and the posterior superior iliac spine and the horizontal line.

FN: The difference between the center of the eyebrows and the coronal plane is positive in forward and negative in backward.

FUPB: The difference between the midpoint of the left and right acromion and the coronal plane is positive in forward and negative in backward.

FLPB: The difference between the midpoint of the left and right greater trochanter and the coronal plane is positive in forward and negative in backward.

NT: On the horizontal plane, the difference between the left and right ear foramen and the central axis is negative in the left posterior torsion and positive in the right posterior torsion.

ST: On the horizontal plane, the difference between the left and right acromion and the central axis is negative in the left posterior torsion and positive in the right posterior torsion.

WT: On the horizontal plane, the difference between the left greater trochanter and the central axis of the greater trochanter was negative in the left posterior torsion and positive in the right posterior torsion.

SL: The horizontal position of the acromion is negative to the left and positive to the right.

WL: The horizontal position of the anterior superior iliac spine is negative on the left and positive on the right.

LNI: The difference between the center of the eyebrows and the sagittal plane is negative to the left and positive to the right.

LUPBI: The difference between the sternal notch and the sagittal plane is negative to the left and positive to the right.

LLPBI: The difference between the navel and the sagittal plane is negative to the left and positive to the right.

It is well known that spinal curvature is highly correlated with age (*Youdas, Hollman & Krause, 2006*; *Lang-Tapia et al., 2011*). Therefore, in order to minimize the interference caused by age, we analyzed the data in the SG and the CG at the same age. The age was obtained from the survey, which was the year of age from the participants' birthday plus the days from the birthday to the date of completion of the first questionnaire survey, converted to years.

## Statistical analysis

IBM SPSS *v.* 25.0 (IBM Corp., Chicago, IL, USA) was used to analyze the data. The average and standard deviation were reported for all variables, unless otherwise stated. Shapiro–Wilk test was used to assess the normality of the variables in different SGs and CGs. Independent-samples *t*-test was performed to compare two groups' means for each variable that demonstrated normal distribution, while Mann–Whitney *U* test was used for the variables that did not confirm to normal distribution. Kappa statistics was used to determine the level of reliability of the data collected from the questionnaire, and ICC was used for the reliability of the postural assessment. The significance level was set to $p \leq 0.05$.

## RESULTS

The repeated completion of the questionnaire showed that the self-reported training time had a high Cohen's kappa coefficient (Cohen's kappa = 0.93, $p = 0.005$) for the first two completions, as well as for the first *vs* third completions (Cohen's kappa = 0.87, $p = 0.003$), demonstrating a high level of consistency. The repeated completion of the postural assessment showed that the postural assessment had a high reliability (ICC = 0.91, $p = 0.005$).

The age range and basic demographic data of each SG are shown in Table 1. The results show that there were no significant differences in height, weight and Body Mass Index (BMI) between the SG and the CG at the same age.

The statistical methods used by each variable are shown in Tables 1 and 2. The comparison of the body posture assessments between the SGs and the CG are shown in Table 2. The NSTKA ($t = -2.560$; $p = 0.011$) of the swimming group were significantly larger, and the NSLLA ($t = 0.344$; $p = 0.024$) and USLLA ($t = 5.738$; $p = 0.030$) was significantly smaller, compared with the CG. The results detected no significant difference in the body posture in the rope skipping group and the CG. The results show that the USTKA ($t = -1.970$; $p = 0.050$) of the running group was significantly larger. The results detected no significant difference in the body posture between the basketball group and the CG. The results detected no significant difference in the body posture between the badminton group and the CG. The results showed that the USTKA ($t = 2.096$; $p = 0.038$) and DLTKA ($t = 3.076$; $p = 0.003$) of the table tennis group was significantly lower, the ST ($t = 2.389$; $p = 0.019$) and WT ($t = 2.511$; $p = 0.013$) was significantly smaller compared with the CG. The NSTKA ($t = -2.754$; $p = 0.006$) of the football group were significantly higher compared with the CG.

## DISCUSSION

There have been studies in the literature that have shown significant correlations between sex, age, height, weight and variables of body posture (*Arshad et al., 2019*; *Intolo et al., 2009*). In this study, there was no significant difference in age, height, weight and BMI between each sport group and its CG.

Huang et al. (2022), *PeerJ*, DOI 10.7717/peerj.14520
**Table 1** The age range and basic demographic data of the sport and control groups of the same age.

| Variable | Swimming | | Rope skipping | | Running | | Basketball | |
|---|---|---|---|---|---|---|---|---|
| | SG ($n = 104$) | CG ($n = 62$) | SG ($n = 122$) | CG ($n = 62$) | SG ($n = 325$) | CG ($n = 62$) | SG ($n = 139$) | CG ($n = 137$) |
| Age range (y) | 6∼13 | | 6∼13 | | 6∼13 | | 6∼17 | |
| Age (y) | 9.7 ± 1.8 | 9.5 ± 2.5[*] | 9.7 ± 2.3 | 9.5 ± 2.5[*] | 9.9 ± 2.4 | 9.5 ± 2.5[*] | 12.6 ± 2.8 | 12.5 ± 3.3[*] |
| Height (cm) | 142.4 ± 11.6 | 139.5 ± 16.1[*] | 141.7 ± 15.4 | 139.5 ± 16.1[**] | 142.8 ± 14.4 | 139.5 ± 16.1[*] | 157.8 ± 14.6 | 153.5 ± 17.6[*] |
| Weight (kg) | 40.2 ± 14.3 | 37.7 ± 15.7[*] | 39.8 ± 16.7 | 37.7 ± 15.7[*] | 39.2 ± 13.5 | 37.7 ± 15.7[*] | 54.3 ± 19.2 | 51.5 ± 20.6[*] |
| BMI (kg m$^{-2}$) | 19.4 ± 4.9 | 18.5 ± 3.9[*] | 19.0 ± 4.5 | 18.5 ± 3.9[*] | 18.7 ± 3.7 | 18.5 ± 3.9[*] | 21.4 ± 5.0 | 20.8 ± 5.5[*] |

| Variable | Badminton | | Table tennis | | Football | |
|---|---|---|---|---|---|---|
| | SG ($n = 116$) | CG ($n = 110$) | SG ($n = 25$) | CG ($n = 91$) | SG ($n = 96$) | CG ($n = 137$) |
| Age range (y) | 9∼17 | | 13∼19 | | 6∼15 | |
| Age (y) | 14.3 ± 2.1 | 13.8 ± 2.1[*] | 14.3 ± 1.8 | 14.6 ± 1.2[*] | 11.8 ± 2.6 | 12.5 ± 3.3[*] |
| Height (cm) | 162.2 ± 10.7 | 160.3 ± 11.6[*] | 162.8 ± 9.7 | 164.3 ± 7.4[*] | 154.2 ± 16.3 | 153.5 ± 17.6[*] |
| Weight (kg) | 57.1 ± 17.6 | 57.3 ± 18.2[*] | 56.5 ± 18.4 | 61.1 ± 16.9[*] | 52.9 ± 21.9 | 51.1 ± 20.6[*] |
| BMI (kg m$^{-2}$) | 21.3 ± 4.9 | 21.9 ± 5.5[*] | 21.0 ± 5.0 | 22.5 ± 5.6[*] | 21.3 ± 5.5 | 20.8 ± 5.5[*] |

**Notes.**

SG, sport group; CG, control group; BMI, Body Mass Index.

[*]*p*-values were determined from Mann–Whitney *U* test.

[**]*p*-values were determined from independent-samples *t*-test; The age was displayed by both the age range of the participants and the mean ± SD of the group. No significant difference was found between SG and CG in all groups.

**Table 2 Comparisons of body posture assessment results between students participating in specific sports and their age-matched controls.**

| Variable | Swimming | | | Rope skipping | | | Running | | |
|---|---|---|---|---|---|---|---|---|---|
| | SG (n = 104) | CG (n = 62) | t | SG (n = 112) | CG (n = 62) | t | SG (n = 325) | CG (n = 62) | t |
| NI (°) | 89.0 ± 12.1 | 88.8 ± 15.6**** | −0.06 | 89.4 ± 14.1 | 88.8 ± 15.6*** | −0.272 | 88.9 ± 15.5 | 88.8 ± 15.6**** | −0.033 |
| NSTKA (°) | 35.0 ± 9.5 | 31.2 ± 8.5***** | −2.560 | 33.5 ± 8.6 | 31.2 ± 8.5*** | −1.686 | 33.3 ± 9.2 | 31.2 ± 8.5**** | −1.653 |
| NSLLA (°) | −23.0 ± 11.0 | −27.0 ± 11.1**** | −2.278 | −26.8 ± 10.8 | −27.0 ± 11.1*** | −0.144 | −25.3 ± 11.9 | −27.0 ± 11.1**** | −1.021 |
| PI (°) | 24.6 ± 6.24 | 23.2 ± 7.8*** | −1.267 | 23.4 ± 7.2 | 23.2 ± 7.8*** | −0.185 | 22.7 ± 7.3 | 23.2 ± 7.8**** | 0.468 |
| USTKA (°) | 25.1 ± 9.4 | 22.6 ± 9.9*** | −1.628 | 25.1 ± 9.6 | 22.6 ± 9.9*** | −1.655 | 25.1 ± 9.3 | 22.6 ± 9.9***** | −1.970 |
| USLLA (°) | −11.7 ± 8.4 | −15.2 ± 12.3***** | −1.987 | −13.1 ± 8.9 | −15.2 ± 12.3*** | −1.170 | −13.3 ± 8.3 | −15.2 ± 12.3**** | −1.183 |
| LTKA (°) | 32.0 ± 9.8 | 30.4 ± 11.3**** | −0.931 | 32.3 ± 10.1 | 30.4 ± 11.3**** | −1.119 | 31.9 ± 9.4 | 30.4 ± 11.3*** | −0.984 |
| LLLA (°) | 3.3 ± 12.6 | 0.8 ± 12.4*** | −1.204 | 1.2 ± 12.9 | 0.8 ± 12.4**** | −0.188 | 1.8 ± 13.1 | 0.8 ± 12.4**** | −0.530 |
| FN (cm) | 2.5 ± 2.6 | 2.3 ± 2.7*** | −0.267 | 1.8 ± 3.1 | 2.3 ± 2.7*** | 1.229 | 2.1 ± 2.8 | 2.3 ± 2.7**** | 0.645 |
| FUPB (cm) | 3.0 ± 3.3 | 3.3 ± 3.0*** | 0.607 | 3.5 ± 2.7 | 3.3 ± 3.0*** | −0.565 | 3.3 ± 2.9 | 3.3 ± 3.0**** | −0.089 |
| FLPB (cm) | 1.9 ± 3.7 | 2.4 ± 3.6*** | 0.788 | 2.0 ± 3.4 | 2.4 ± 3.6*** | 0.738 | 2.0 ± 3.5 | 2.4 ± 3.6**** | 0.867 |
| NT (cm) | −0.2 ± 4.1 | −0.8 ± 4.3*** | −1.019 | −0.7 ± 4.2 | −0.8 ± 4.3*** | −0.228 | −0.3 ± 4.3 | −0.8 ± 4.3**** | −0.956 |
| ST (cm) | −0.9 ± 3.6 | −0.8 ± 3.6*** | 0.063 | −0.7 ± 3.4 | −0.8 ± 3.6*** | −0.263 | −0.6 ± 3.6 | −0.8 ± 3.6**** | −0.440 |
| WT (cm) | −1.3 ± 2.4 | −1.3 ± 2.3*** | −0.056 | −1.1 ± 2.3 | −1.3 ± 2.3*** | −0.564 | −1.1 ± 2.4 | −1.3 ± 2.3**** | −0.498 |
| SL (cm) | 0.2 ± 2.4 | −0.1 ± 2.4*** | −0.699 | −0.2 ± 2.4 | −0.1 ± 2.4*** | 0.244 | −0.1 ± 2.6 | −0.1 ± 2.4**** | 0.111 |
| WL (cm) | −0.5 ± 1.5 | −0.3 ± 1.3*** | 0.996 | −0.6 ± 1.5 | −0.3 ± 1.3*** | 1.171 | −0.3 ± 1.4 | −0.3 ± 1.3**** | 0.119 |
| LNI (cm) | −1.0 ± 3.7 | −0.4 ± 3.8*** | 0.901 | −1.4 ± 3.8 | −0.4 ± 3.8*** | 1.594 | −1.2 ± 3.6 | −0.4 ± 3.8**** | 1.487 |
| LUPBI (cm) | −1.3 ± 3.3 | −0.7 ± 3.4*** | 1.051 | −1.2 ± 3.3 | −0.7 ± 3.4*** | 0.953 | −1.3 ± 3.4 | −0.7 ± 3.4**** | 1.170 |
| LLPBI (cm) | −1.4 ± 3.0 | −0.7 ± 3.2*** | 1.472 | −1.5 ± 3.0 | −0.7 ± 3.2*** | 1.553 | −1.4 ± 3.1 | −0.7 ± 3.2**** | 1.670 |

| Variable | Basketball | | | Badminton | | | Table tennis | | |
|---|---|---|---|---|---|---|---|---|---|
| | SG (n = 139) | CG (n = 137) | t | SG (n = 116) | CG (n = 110) | t | SG (n = 25) | CG (n = 91) | t |
| NI (°) | 80.7 ± 13.1 | 82.5 ± 14.6*** | 1.121 | 79.9 ± 12.4 | 80.0 ± 13.9*** | 1.379 | 76.4 ± 12.2 | 77.0 ± 11.3*** | 1.966 |
| NSTKA (°) | 33.0 ± 7.2 | 31.9 ± 7.5**** | −1.085 | 31.7 ± 6.7 | 32.4 ± 7.4*** | 0.102 | 30.0 ± 7.2 | 32.1 ± 6.8**** | 1.121 |
| NSLLA (°) | −27.4 ± 9.4 | −27.1 ± 10.1**** | 0.256 | −25.9 ± 9.2 | −27.3 ± 9.1*** | −1.012 | −28.9 ± 9.0 | −27.2 ± 8.8**** | 0.837 |
| PI (°) | 20.0 ± 5.7 | 21.1 ± 6.7*** | 1.554 | 19.0 ± 5.1 | 19.5 ± 5.0*** | 2.876 | 20.0 ± 5.7 | 19.2 ± 4.8*** | 0.782 |
| USTKA (°) | 25.7 ± 8.5 | 25.1 ± 9.1**** | −0.508 | 25.1 ± 7.4 | 26.3 ± 8.3*** | 0.126 | 23.3 ± 8.9 | 27.0 ± 7.6***** | 0.936 |
| USLLA (°) | −13.0 ± 9.7 | −13.2 ± 10.6*** | −0.197 | −11.1 ± 8.8 | −12.6 ± 9.9**** | −1.840 | −12.4 ± 8.6 | −11.9 ± 8.7**** | −0.359 |
| LTKA (°) | 35.2 ± 9.8 | 34.4 ± 10.5**** | −0.695 | 35.2 ± 8.8 | 36.5 ± 9.5*** | −0.691 | 31.0 ± 8.6 | 37.1 ± 8.9****** | 1.516 |
| LLLA (°) | 0.0 ± 13.4 | −0.3 ± 12.1*** | −0.153 | −0.7 ± 10.5 | −1.1 ± 11.7**** | 0.246 | −1.5 ± 12.8 | −1.5 ± 11.9**** | 0.463 |
| FN (cm) | 2.2 ± 2.6 | 2.4 ± 2.6*** | 0.475 | 2.0 ± 2.7 | 2.4 ± 2.6*** | 1.098 | 1.9 ± 2.5 | 2.3 ± 2.5*** | 0.836 |
| FUPB (cm) | 2.8 ± 3.5 | 3.3 ± 3.0*** | 1.392 | 3.0 ± 3.3 | 3.3 ± 3.1*** | 0.777 | 3.2 ± 3.0 | 3.2 ± 3.2*** | 0.228 |
| FLPB (cm) | 1.5 ± 3.9 | 2.2 ± 3.6*** | 1.396 | 1.8 ± 3.7 | 1.9 ± 3.8*** | 0.850 | 1.1 ± 4.0 | 1.9 ± 3.7*** | 1.373 |
| NT (cm) | −0.5 ± 4.4 | −0.4 ± 4.4*** | 0.276 | −0.1 ± 4.2 | −0.1 ± 4.4*** | −0.557 | −1.0 ± 4.7 | −0.2 ± 4.4*** | 0.598 |
| ST (cm) | −0.4 ± 3.6 | −0.7 ± 3.7*** | −0.669 | −0.2 ± 3.7 | −0.4 ± 3.7*** | −0.936 | −0.5 ± 3.8 | −0.7 ± 3.7*** | −0.275 |
| WT (cm) | −1.1 ± 2.5 | −1.5 ± 2.3*** | −1.336 | −1.0 ± 2.6 | −1.5 ± 2.4*** | −1.522 | −0.8 ± 2.3 | −1.6 ± 2.3*** | −1.229 |
| SL (cm) | −0.4 ± 2.9 | 0.0 ± 2.5*** | 1.410 | −0.2 ± 2.6 | 0.2 ± 2.5*** | 0.701 | −1.3 ± 2.1 | 0.0 ± 2.5*** | 2.498 |
| WL (cm) | −0.5 ± 1.6 | −0.3 ± 1.5*** | 1.173 | −0.4 ± 1.6 | −0.3 ± 1.6*** | 0.584 | −1.2 ± 1.7 | −0.3 ± 1.7**** | 2.873 |
| LNI (cm) | −1.1 ± 3.7 | −1.1 ± 3.8*** | −0.019 | −0.9 ± 3.6 | −1.1 ± 3.8*** | −0.635 | −2.4 ± 3.4 | −1.6 ± 3.7*** | 1.586 |

**Table 2** (*continued*)

| Variable | Basketball | | | Badminton | | | Table tennis | | |
| --- | --- | --- | --- | --- | --- | --- | --- | --- | --- |
| | SG (*n* = 139) | CG (*n* = 137) | *t* | SG (*n* = 116) | CG (*n* = 110) | *t* | SG (*n* = 25) | CG(*n* = 91) | *t* |
| LUPBI (cm) | −1.2 ± 3.4 | −1.3 ± 3.4*** | −0.346 | −1.2 ± 3.4 | −1.3 ± 3.4*** | −0.337 | −1.8 ± 3.2 | −1.7 ± 3.3*** | 0.674 |
| LLPBI (cm) | −1.3 ± 3.1 | −1.3 ± 3.2*** | 0.065 | −2.0 ± 2.9 | −1.3 ± 3.2*** | 1.715 | −1.1 ± 3.4 | −1.7 ± 3.1*** | −0.315 |

| Variable | Football | | |
| --- | --- | --- | --- |
| | SG (*n* = 96) | CG (*n* = 137) | t |
| NI (°) | 85.1 ± 16.1 | 82.5 ± 14.6*** | −1.273 |
| NSTKA (°) | 34.6 ± 7.2 | 31.9 ± 7.5***** | −2.754 |
| NSLLA (°) | −26.6 ± 9.1 | −27.1 ± 10.1**** | −0.400 |
| PI (°) | 20.7 ± 5.3 | 21.1 ± 6.7*** | 0.491 |
| USTKA (°) | 27.5 ± 9.2 | 25.1 ± 9.1*** | −1.943 |
| USLLA (°) | −10.7 ± 9.3 | −13.2 ± 10.6**** | −1.917 |
| LTKA (°) | 35.9 ± 8.7 | 34.3 ± 10.5*** | −1.206 |
| LLLA (°) | 1.7 ± 12.2 | −0.3 ± 12.1**** | −1.186 |
| FN (cm) | 2.6 ± 2.5 | 2.4 ± 2.6*** | −0.808 |
| FUPB (cm) | 3.4 ± 2.9 | 3.3 ± 3.0*** | −0.121 |
| FLPB (cm) | 2.4 ± 3.4 | 2.2 ± 3.6*** | −0.398 |
| NT (cm) | −0.2 ± 4.4 | −0.4 ± 4.4*** | −0.271 |
| ST (cm) | −0.7 ± 3.5 | −0.7 ± 3.7*** | −0.006 |
| WT (cm) | −1.1 ± 2.3 | −1.5 ± 2.3*** | −1.136 |
| SL (cm) | −0.4 ± 2.6 | 0.0 ± 2.5*** | 1.274 |
| WL (cm) | −0.2 ± 1.5 | −0.3 ± 1.5*** | −0.152 |
| LNI (cm) | −1.0 ± 3.9 | −1.1 ± 3.8*** | −0.133 |
| LUPBI (cm) | −0.8 ± 3.4 | −1.3 ± 3.4*** | −1.160 |
| LLPBI (cm) | −1.4 ± 3.1 | −1.3 ± 3.2*** | 0.232 |

**Notes.**

SG, sport group; CG, control group.

*$p < 0.05$.

**$p < 0.01$.

***$p$-values were determined from Mann–Whitney $U$ test.

****$p$-values were determined from independent-samples $t$-test.

NI, neck inclination; NSTKA, natural standing thoracic kyphosis angle; NSLLA, natural standing lumbar lordosis angle; PI, pelvic inclination; USTKA, upright sitting thoracic kyphosis angle; USLLA, upright sitting lumbar lordosis angle; LTKA, learning thoracic kyphosis angle; LLLA, learning lumbar lordosis angle; FN, forward neck; FUPB, forward upper part of body; FLPB, forward lower part of the body; NT, neck torsion; ST, shoulder torsion; WT, waist torsion; SL, shoulder level; WL, waist level; LNI, lateral neck inclination; LUPBI, lateral upper part of body inclination; LLPBI, lateral lower part of body inclination.

Compared with mature bones, teenagers' bones contain more organic matter, less inorganic matter, and demonstrate a greater adaptability to physical loads (*Carter, 1984*; *Hou et al., 1990*), therefore are more likely to be affected by sports training. Our results showed that, although the sports training engaged by these students was relatively lower compared with professional sports training, it could still have an impact on the physical posture of teenagers. Such effects can be considered as either a treatment or correction (*Carman et al., 1985*), or a pathogenic factor for anomalies in body posture (*Tanchev et al., 2000*).

Our results showed that compared with the CG, the NSTKA of the swimming group were significantly larger, and the NSLLA and USLLA was significantly smaller. During swimming, swimmers are in a prone position in the water and mainly use the anterior chain muscles, including chest, serratus anterior and rectus abdominis, to generate force

in water to push the body forward (*Peterson et al., 1997*). The tension of these muscles is increased as a result of swimming exercise, so that the body is easily forming a forward flexion. Our findings are similar to that of *Zaina et al. (2015)* who compared the ratio of abnormal TKA and LLA between a swimming group and a control group and reported that the risk of hyperkyphosis was 2.32 times higher, and the risk of hyperlordosis was 2.24 times higher in the swimming group than that in the control.

Our results also showed that compared with the CG, the USTKA of the runners was significantly larger. *Lichota, Plandowska & Mil (2011)* reported that volleyball players and runners had larger TKA than taekwondo and handball athletes, but there was no significant difference between the volleyball players and the runners. *Grabara (2015)*, in comparison of spine curvature of volleyball players and blank controls, found that the volleyball players had a larger TKA. At present, there has been no research report on the effect of running on spinal curvature to our knowledge. According to the results of *Lichota, Plandowska & Mil (2011)* and *Grabara (2015)* it is speculated that running could tend to increase TKA, and our study results appear to support this speculation.

The football players in our current study showed a significantly higher NSTKA as compared with the CG. *Grabara (2012)* studied the effect of football on adolescent spinal curvature and reported that, compared with the control group, football players had a significantly higher LLA, but there was no significant difference in TKA. In the study of *Grabara (2012)* there were significant differences in body weight and BMI between the football group and the control. The influence of body weight and BMI may be the reason why the results of *Grabara (2012)* are different from ours. *Watson & Mac Donncha (2000)* studied the relationship between the cumulative sport training hours and the sagittal curvature of the immature spine and reported that, the TKA and LLA became larger when they were engaged in football for 282 h per year, compared with the blank control. In our study, an inclusion criterion of the participants in the football group was that the practicing time was no less than 4 h per week, and the total exercise time of no less than 192 h a year. However, the intensity of amateur participants could be much less than that of professional athletes. The lower exercise intensity and exercise time may be the reason why the LLA did not show a significant difference between the football and the control group in this study.

The table tennis group in our study showed a smaller USTKA and DLTKA, compared with the CG. Table tennis exercise requires athletes to lean forward slightly. In order to maintain this posture, it is necessary to contract the posterior bracelet muscles, whose function is to limit forward flexion and keep the body upright. Long-term table tennis practice may strengthen the posterior bracelet muscles. The enhancement of the posterior chain muscle group may be the reason for the decrease of compensation of TKA in upright sitting and desk learning position in the table tennis group. Table tennis is one of the fastest sports (*Kondrič, Zagatto & Sekulić, 2013*), mainly because the distance between athletes is short and the ball speed is fast, so the reaction time of athletes is generally short. In order to choose the optimal angle and position when hitting the ball, the players usually adopt the preparation posture. *Bańkosz & Barczyk-Pawelec (2020)* studied the posture difference between table tennis players in normal standing and the preparation posture, and showed that when in the preparing posture, the inclination angle of trunk increased. Our research

results also showed that the right shoulder and anterior superior iliac spine of the table tennis group was higher, which may be due to the fact that most teenagers were right handed (*Scharoun & Bryden, 2014*). When in a preparation posture, they would tilt their torso to the left and elevate their right shoulder. Long-term practice with that posture would lead to a higher level of the right shoulder in the table tennis group. Our study showed that asymmetrical movement was more likely to make the body deviate from frontal plane and horizontal plane.

Our study showed that there was no significant difference in body posture between the basketball group and the CG. *Guedes & João (2014)* studied the difference of body posture between young basketball players and non-athletes of the same age, and reported that, compared with the control group, the TKA of basketball players was smaller in the left view, and there was no significant difference in the right view; and there was no significant difference in the left and right views of LLA. In the research of *Guedes & João (2014)* there were significant differences in height and weight between the basketball group and the control, and the influence of height and weight may be the reason for why their results are different from ours.

*Wojtys et al. (2000)* carried out a study similar to this study. *Wojtys et al. (2000)* measured TKA and LLA in 2270 participants, then correlated these data with cumulative training time reported in the questionnaire, and explored the relationship between cumulative sport training time and sagittal curvature of immature spine. The study showed that cumulative sport training time was positively correlated with sagittal curvature of immature spine.

Some studies have shown that asymmetric movement may lead to asymmetric body posture (*Grabara, 2020*). However, in the sports investigated in this study, basketball, football, badminton and table tennis all belong to asymmetric movement. Only table tennis players demonstrated an asymmetrical body posture. This may be related to the increase of torso tilt angle when preparing posture in table tennis, or it may be because basketball, football and badminton need more training intensity and time to cause an asymmetric body posture. Further studies are needed to verify these speculations.

There are limitations in this study. First, a cross-sectional research design was used instead of a longitudinal study, therefore the strength of evidence to prove the cause–effect relationship is relatively low. Secondly, the participants were not randomly recruited from large number of schools in the country, so the application of the research results may not be directly generalized to wider student populations. In future research, it is necessary to include longitudinal training or intervention, and randomly recruit participants from different regions and schools to further determine the influence of sport participation, amount of exercise, length of training and other factors on teenagers' body posture and its underlying mechanism. Finally, the study was to compare between the sports participation group and the control group. The two groups included any participants who meet the inclusion criteria, regardless of their age. Although there could be an effect from normal growth and development in the age span of these participants, it was not possible to divide participants into groups in smaller age ranges due to the number of participants in each age range. Nevertheless, there was no significant difference in the average age between the two groups. A larger number of participants should be recruited in future studies to investigate

the effect of chronological or biological age (*e.g.*, in primary and high school students, or before and during puberty).

## CONCLUSIONS

The results of this study showed that long-term participation in some popular amateur sports had a significant and specific relationship on teenagers' body posture, such as in natural standing, upright sitting and desk learning postures.

Teenagers who had a long-term participation in certain sports practice, including swimming, football and running, demonstrated a bending forward posture. The participants of table tennis showed less spine curvature in the sagittal plane, and body deviation from the frontal plane and horizontal plane.

It is recommendable for teenagers to monitor their posture regularly when practicing in sports, for early detection of posture changes, and for evaluating the impact of such changes on musculoskeletal health.

## ACKNOWLEDGEMENTS

The authors would like to thank the participating students, parents/guardians and teachers for their dedicated collaboration.

### Funding

This research was funded by the National Natural Science Foundation of China, grant number 31271275. The funders had no role in study design, data collection and analysis, decision to publish, or preparation of the manuscript.

### Grant Disclosures

The following grant information was disclosed by the authors:
National Natural Science Foundation of China: 31271275.

### Competing Interests

The authors declare there are no competing interests.

### Author Contributions

- Yongchao Huang performed the experiments, analyzed the data, prepared figures and/or tables, authored or reviewed drafts of the article, and approved the final draft.
- Meiling Zhai performed the experiments, analyzed the data, prepared figures and/or tables, authored or reviewed drafts of the article, and approved the final draft.
- Shi Zhou analyzed the data, authored or reviewed drafts of the article, and approved the final draft.
- Yahong Jin conceived and designed the experiments, authored or reviewed drafts of the article, and approved the final draft.

- Li Wen conceived and designed the experiments, authored or reviewed drafts of the article, and approved the final draft.
- Yuqi Zhao performed the experiments, prepared figures and/or tables, authored or reviewed drafts of the article, and approved the final draft.
- Xu Han performed the experiments, authored or reviewed drafts of the article, and approved the final draft.

## Human Ethics

The following information was supplied relating to ethical approvals (*i.e.*, approving body and any reference numbers):

The study was approved by the Ethics Committee of Tianjin University of Sports (approval number1 TJUS2019032).

## Data Availability

The raw data is available in the Supplemental Files.

## Supplemental Information

Supplemental information for this article can be found online at http://dx.doi.org/10.7717/peerj.14520#supplemental-information.

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
