# Peer review of "Influence of long-term participation in amateur sports on physical posture of teenagers"

_PeerJ, doi:10.7717/peerj.14520_

## Round 0.1 · original submission · Major Revisions

Thank you for submitting the manuscript to PeerJ. It has been reviewed by experts in the field and we request that you make major revisions before it is processed further.

We look forward to hearing from you soon.

Best wishes,

Badicu Georgian, Ph.D

·

Basic reporting

Influence of long-term participation in recreational sports on physical posture of teenagers (#73819)
Suggest team of author to change topic in one part.
Influence of long-term participation in competitive/amateurs sports on physical posture of teenagers
Or
Influence of long-term participation in sports on physical posture of teenagers

Participation in recreational sports is a wrong information. In the part with results, they explain that children/students and teenagers have:
The inclusion criteria were: the time of self-practice or training under the guidance of club coach was not less than 4 hours per week and the training history was not less than two years. If the participant was only engaged in one sport for long-term, participant was assigned to that specific SG. If the participants took part in a certain sport for less than one hour every week and the training length was less than one year, they were assigned to the CG.

when they were engaged in football for 282 hours per year, compared with the blank CG 27. In our study, an inclusion criteria of the participants in the football group was that the practicing time was no less than 4 hours per week, and the total exercise time of 192 hours a year.

However, the intensity of amateur participants could be much less than that of professional
athletes.
Experimental groups are a young athlete in competitive sport.
Recreation belongs to grass root sport, massive sport, Sport for all. Basic difference is reasons for activities. People looking for physical activity in free time (after work obligation) to resolve the problem with the stress, nervosas, to “recharge battery” for tomorrow and to have prevention and support for health.
Children and teenagers in research are a young athletes complete concentrate and focused in sports results.

Experimental design

In the parts Materials & Methods we have Participants, Procedure, Statistical analysis, but we don’t have basic information about general Methods

Validity of the findings

201-202
Table 2. The NSTKA (t=-2.560 (df=164);p=0.011) of the swimming group were significantly
larger, and the NSLLA (t=0.344 (df=164);p=0.024) and USLLA (t=5.738 (df=164);p=0.030)
was significantly smaller, compared with the CG.

1. In Table 2. we don’t have t values
2. It is to many dates in on table without t-test values
3. In independent-samples t test df don’t have any role

·

Basic reporting

Why did you make a group so different in age? There are many differences in all areas: anthropometric, motor, psychomotor, psychological, among 6-year-olds and 17-year-olds.
The participants stated that two primary schools and one middle school participated, 17-year-old students are from high school ??
The anthropometric development of the 2 age categories in Recreational Sports are very different…

Experimental design

Is necessary a clear formulation of the study hypothesis.
In the discussions I propose a clearer presentation of the results
I propose a clearer statement of their conclusions.

Validity of the findings

Intervention programs for the sports disciplines listed to be described.
What did the students do in the 192 hours of recreational activities ?

Additional comments

In the part of theoretical substantiation and in the part of discussions I propose several references to authors who have dealt with similar research.
In the bibliography I would propose to add some newer authors after 2018.

Reviewer 3 ·

Basic reporting

Many instances of awkward sentence structure. For example:
Line 221: "and are greater adaptability to physical loads..."
Line 238: "in comparison of spine curvature of volleyball players..."
Line 277: "basketball players and non-athletes of the same age and resulted that..."
Line 299: "Participants in asymmetrical movement was..."
Line 302: "for evaluation the impact..."

Experimental design

Abstract states 1658 students, while the manuscript states 1124 students and does not specify the number excluded.

The 'Materials & Methods' section does not specify the ages or age range of children in the text. This is left for Table 1, which does not make sense as there is listed an age as well as an age range. The title of Table 1 specifies 'sports and controls groups of the same age' but it is difficult to interpret how this is displayed in the Table. Same goes for Table 2.

Statistical analysis accounted for non-normal distributions but authors did not specify how they statistically reduced the chance of a type 1 error by performing so many statistical tests (e.g. Bonferroni correction).

The largest deficiency is the apparent collapse of wide ages ranges (i.e. 6-13 yrs) into categories to compare control and sport groups. Obviously, the developmental changes across this age span are huge and such comparisons represent a flawed design.

Validity of the findings

The validity of the findings are suspect due to the deficiencies outlined above.

---

## Round 0.2 · Minor Revisions

Abstract - Results: please add some numerical data obtained in this study.
From statistical analysis IBM SPSS (ver. 25) please add if you used "IBM Corp., Chicago, IL, USA".

Tables 1 and 2 look very ugly on the page and are not very easy to understand, please redo them.

·

Basic reporting

Authors accept all suggestions. New version is correct.

Experimental design

Correct.

Validity of the findings

New version is correct.

·

Basic reporting

Yes is clear and unambiguous, professional English used throughout.

Literature references, sufficient field background/context provided is better now.

Figures are relevant to the content of the article with sufficient resolution, and appropriately described and labeled.

The submission include all results relevant to the hypothesis.

Experimental design

This paper is an original primary research within Aims and Scope of the journal.

The submission is clearly define the research question, which is relevant and meaningful.

The investigation is conducted rigorously and to a high technical standard. The research is conducted in conformity with the prevailing ethical standards in the field.

The research must have been conducted in conformity with the prevailing ethical standards in the field.

Methods described with sufficient detail & information to replicate.

Validity of the findings

Decisions is made based on any subjective determination of impact, degree of advance, novelty or being of interest to only a niche audience. We will also consider studies with null findings.

Replication studies will be considered provided the rationale for the replication, and how it adds value to the literature, is clearly described.

The data are robust, statistically sound, and controlled.

The conclusions are appropriately stated, connected to the original question investigated, and should be limited to those supported by the results. In particular, claims of a causative relationship should be supported by a well-controlled experimental intervention. Correlation is not causation.

Additional comments

Now the article is better. The authors of the article improve all the request which we ask.

---

## Round 0.3 · accepted · Accept

Your manuscript has been accepted for publication. Congratulations!

PeerJ - Life and Environment
Academic Editor
peerj.com/BadicuGeorgian